Identification of the whole genome of alternative splicing and RNA-binding proteins involved in nintedanib-induced apoptosis in gastric cancer cells

Dong Xiaohua 1 2 3
Liu Zhilong 4
Yu Miao 2 3
Yang Xiaojun 1 2 3
Cai Hui caialonteam@163.com 1 2 3
1 The First School of Clinical Medicine, Lanzhou University , LanZhou , Gansu , China
2 Key Laboratory of Molecular Diagnostics and Precision Medicine for Surgical Oncology in Gansu Province, Gansu Provincial Hospital , LanZhou , Gansu , China
3 NHC Key Laboratory of Diagnosis and Therapy of Gastrointestinal Tumor, Gansu Provincial Hospital , LanZhou , Gansu , China
4 Department of Anesthesiology, Gansu Provincial Hospital , LanZhou , Gansu , China
Korla Praveen
Electronic publication date: 2024 Dec 23
Publication date: 2024
Volume: 12
Electronic Location ID: e18697
Received 2024 Jul 5; Accepted 2024 Nov 21
Copyright: ©2024 Dong et al.
Copyright year: 2024
Copyright holder: Dong et al.
License: This is an open access article distributed under the terms of the Creative Commons Attribution License, which permits unrestricted use, distribution, reproduction and adaptation in any medium and for any purpose provided that it is properly attributed. For attribution, the original author(s), title, publication source (PeerJ) and either DOI or URL of the article must be cited.
License URL: https://creativecommons.org/licenses/by/4.0/

Keywords: Gastric cancer, RNA-sequencing, Alternative splicing, RNA-binding proteins

Funding: The National Natural Science Foundation of China No. 82360498 The Natural Science Foundation of Gansu Province No. 23JRRA1756 The National Health Commission Key Laboratory of Diagnosis and Therapy of Gastrointestinal Tumor 2022 Master/Postdoctoral Fund NHCDP2022015 The 2021 Central-Guided Local Science and Technology Development Fund ZYYDDFFZZJ-1 2024 Gansu Provincial Hospital Special Fund of NHC Key Laboratory of Diagnosis and Therapy of Gastrointestinal Tumor paid for the APC 23GSSYA-10 This work was supported by the National Natural Science Foundation of China (No. 82360498), the Natural Science Foundation of Gansu Province (No. 23JRRA1756); the National Health Commission Key Laboratory of Diagnosis and Therapy of Gastrointestinal Tumor 2022 Master/Postdoctoral Fund (NHCDP2022015), and the 2021 Central-Guided Local Science and Technology Development Fund (ZYYDDFFZZJ-1). 2024 Gansu Provincial Hospital Special Fund of NHC Key Laboratory of Diagnosis and Therapy of Gastrointestinal Tumor (23GSSYA-10) paid for the APC. The funders had no role in study design, data collection and analysis, decision to publish, or preparation of the manuscript.

==============================
Background

It has been demonstrated that nintedanib can inhibit the proliferation of gastric cancer cells, but the specific mechanism of action is unclear.

Objective

Investigating the changes of key factors involved in gene transcription and post-transcriptional regulation during the process of treating gastric cancer with nintedanib.

Methods

In this study, we performed transcriptome sequencing on gastric cancer cell groups treated with nintedanib and control groups. The SUVA (Splice sites Usage Variation Analysis) software was used to identify differential alternative splicing (AS) events between the nintedanib-treated group and the control group. Gene ontology (GO) and Kyoto Encyclopedia of Genes and Genomes (KEGG) pathway analyses were conducted to assess the functional differences and pathways associated with these events. Finally, a co-expression regulatory network of differentially expressed RNA-binding proteins (RBPs) and differentially spliced genes was established. Results: A total of 915 differential AS events were identified between the two groups, and these differential genes were closely related to the apoptosis pathway. Further analysis revealed that differential RBPs (TAGLN2, TAGLN, SRSF6, PKM, SRSF2, NOC2L, IPO4, C1QBP, DHX9) may affect the anti-proliferative effect of nintedanib on gastric cancer cells by regulating downstream genes involved in cell proliferation and angiogenesis (NR4A1, BBC3, IFI27) through alternative splicing.

Conclusion

This study systematically identified important changes in alternative splicing and RNA-binding proteins during the process of nintedanib-induced apoptosis in gastric cancer cells. It innovatively revealed the mechanisms of action of nintedanib in gastric cancer cells and expanded the selection of new targets for gastric cancer treatment.

Introduction

Gastric cancer (GC) is a heterogeneous malignant disease associated with environmental and genetic predisposing factors. It is one of the most common malignant tumors in the digestive system and has a high mortality rate (Sung et al., 2021). Due to its insidious onset and propensity for metastasis, a significant number of patients are diagnosed at advanced stages (Eom et al., 2022). Despite significant advancements in surgical techniques, chemotherapy regimens, immunotherapy, and neoadjuvant therapies over the past few decades, the 5-year survival rate for gastric cancer remains around 20% worldwide (Sato et al., 2023). The treatment of gastric cancer continues to pose a challenge for healthcare professionals and researchers worldwide.

Tyrosine kinase inhibitors (TKIs) are one of the most extensively studied targeted therapeutic drugs for cancer treatment. Nintedanib is a triple angiokinase inhibitor that can selectively inhibit the generation of angiogenesis signals mediated by VEGFR, FGFR, and PDGFR (Awasthi & Schwarz, 2015). In in vivo experimental models, nintedanib has shown promising anti-angiogenic and anti-tumor effects (Hilberg et al., 2008). Studies have demonstrated that nintedanib inhibits STAT3 phosphorylation and upregulates Beclin1 to suppress tumor growth in gastric cancer cell lines (Zhu et al., 2023). In the treatment of gastric adenocarcinoma, nintedanib can attenuate tumor cell proliferation, reduce tumor vascular system growth, increase tumor cell death, and promote apoptosis (Awasthi et al., 2023). Nintedanib, in combination with blockade of VEGFR1-3, PDGF α/ β, and FGFR1-3, has shown favorable therapeutic efficacy in the treatment of metastatic gastric esophageal cancer (Won et al., 2019).

RNA binding proteins (RBPs) are proteins that can regulate gene expression and function by binding to RNA. During RNA transcription and post-transcriptional processes such as alternative splicing, modification, transport, translation, and degradation, various specific RBPs bind to RNA and regulate transcription and post-transcriptional processes. RBPs are abnormally deregulated in many human cancers, including gastric cancer, affecting the function of mRNA encoding proteins and contributing to carcinogenesis (El-Rifai et al., 2002). Abnormal expression or loss of function of RBPs also plays a role in related diseases of gastric cancer and represents potential therapeutic targets (Masuda & Kuwano, 2019). However, the role of RBPs in the process of drug action on gastric cancer cells still needs further research to elucidate.

Alternative splicing is a driving factor for post-transcriptional variation, which rearranges introns and exons of immature pre-mRNA transcripts in different ways than standard splicing (Koch, 2017). Deep sequencing methods and new bioinformatics algorithms have provided new insights into the prevalence of splice variants, tissue-specific splicing patterns, and the significance of selective splicing in development and disease. The role of RBPs in regulating alternative splicing (AS) has also attracted significant attention from scientists. Currently, RBPs play important roles in the occurrence and development of AS, including in gastric cancer research (Jun et al., 2022; Masuda & Kuwano, 2019; Wang et al., 2021). This study hypothesizes that RBPs may play an important role in the regulation of post-transcriptional processes involving gene transcription during nintedanib treatment for gastric cancer, potentially influencing the occurrence of variable splicing changes in important genes and exerting therapeutic effects.

Materials and Methods

Cell culture

The human gastric cancer cell lines MKN-28 (Shanghai Fuheng Biotechnology Co., Ltd.) and AGS (Procell Life Science & Technology Co., Ltd) were maintained in 1640 medium supplemented with 10% fetal bovine serum and 1% penicillin-streptomycin. The cells were cultured at 37 °C in a 5% CO2 environment.

Cell proliferation assay

Cell proliferation assay was performed using the CCK-8 kit (APExBIO, Houston, TX, USA). According to the manufacturer’s instructions, cells were seeded at 6000 cells per well in a 96-well plate. After incubating at 37 °C for 24 h, different concentrations of nintedanib (0, 0.01, 0.1, 1, 10, and 100 μM) were added to gastric cancer cells. Then, 10 μL of the CCK-8 reagent was added to each well at 24, 48, and 72 h after treatment, followed by incubation at 37 °C for 2 h. The optical density (OD) value was measured at 450 nm wavelength using a microplate reader.

EdU assay

The EdU cell proliferation assay was performed using the EdU kit (MeilunBio; EdU-555, China). MKN-28 and AGS cells were seeded at a density of 5 ×104 cells per well in a 24-well plate. After treating with different concentrations of nintedanib (0, 1, 5, 10 μM) for 48 h, the cells were incubated with 50 μM EdU for 2 h and then fixed with 4% paraformaldehyde for 30 min. After permeabilization with 0.3% Triton X-100 for 15 min and washing with PBS, the cells were incubated with 100 μL of Click reaction mixture in the dark for 30 min. Finally, the cells were incubated for 10 min with Hoechst 33342 and observed under a fluorescence microscope to measure the fluorescence intensity.

Flow cytometry

AGS human gastric cancer cell lines were cultured in RPMI-1640 medium (with 10% FBS) at 37 °C and 5% CO2 until they reached 50–60% confluence. Based on the experimental design, the cells were divided into a control group and treatment groups (nintedanib: 1 µM, 5 µM, 10 µM) and treated for 48 h. The Annexin V-FITC/PI dual staining kit was employed for analysis. A total of 1 × 106 cells were resuspended in 300 µL of PBS, and 5 µL of Annexin V-FITC and 5 µL of PI were added, followed by a 30-minute incubation in the dark. Subsequently, 200 µL of Binding Buffer was added to mix, and flow cytometry analysis was performed immediately. Using the BD FACSCanto II flow cytometer, 30,000 events were recorded, and data were analysed to generate scatter plots distinguishing live cells, early apoptotic cells, late apoptotic cells, and necrotic cells. Results were expressed as percentages, and statistical methods were used to validate the significance of the findings.

RNA-seq

AGS cells were selected for subsequent sequencing analysis. The minimum sample number required for effective data analysis was determined using RNASeqPower (https://rodrigo-arcoverde.shinyapps.io/rnaseq_power_calc/). Estimating that the average depth of coverage from RNA sequencing would be 20 reads per gene, the biological coefficient of variation as 0.2 and alpha was set to 0.05. We determined that each group requires a minimum of 3 replicate samples to detect differential expression, with an effect size of 2 and power of 0.8. Log-phase AGS cells were treated with nintedanib for 48 h, and cellular RNA was extracted. RNA sequencing was performed with the assistance of Genewiz (Azenta Life Sciences, Suzhou, China). A total of 1 µg of total RNA was used for library preparation. Poly(A) mRNA was isolated using Oligo (dT) beads. The mRNA fragments were then subjected to fragmentation under divalent cation and high-temperature conditions. The first-strand cDNA and second-strand cDNA were synthesized using random primers. The purified double-stranded cDNA was then treated to repair the ends, followed by the addition of a dA tail in a single reaction and T-A ligation to add adapters at both ends. Size selection of the adapter-ligated DNA was performed using DNA purification beads. Each sample was PCR amplified using P5 and P7 primers, and the PCR products were validated. Subsequently, libraries with different indices were multiplexed and loaded onto the Illumina HiSeq/Illumina Novaseq/MGI2000 instrument for sequencing, using a 2x150 paired-end (PE) configuration as per the manufacturer’s instructions. The raw data were quality controlled using Cutadapt (V1.9.1) software. Image analysis was performed using HiSeq Control Software version 1.8.4. The DESeq2 Bioconductor package was used for differential expression analysis.

Alternative splicing analysis

To define and quantify alternative splicing (AS) and regulated alternative splicing (RAS) events between the nintedanib-treated group and the control group based on the RNA-seq data, we utilized the SUVA software (Cheng et al., 2021). Statistical analysis was performed on the read counts of all samples, and all detected alternative splicing events (ASEs) were included in the differential analysis (splicing ratio difference is more than 0.15 and P value isless than 0.05). The proportion of events was obtained as the pSAR (percentage of events) using SUVA.

Differentially expressed RBP analysis and RBP–RAS network construction

We screened for all differentially expressed genes between the nintedanib-treated group and the control group (FDR ≤ 0.01, FC ≥ 1.5 or ≤ 1/1.5) and identified an overlap with the existing 2,141 RBP genes. Co-expression analysis was performed for the Differentially expressed RBPs (DE-RBPs) and RAS events (pSAR ≥ 50%) related to cell proliferation and angiogenesis (|Pearson’s correlation| ≥ 0.99, p-value ≤ 0.01), and a regulatory network of RBP–RAS was constructed.

Functional enrichment analysis

To distinguish the functional categories of differentially expressed genes (DEGs), the KEGG Orthology-Based Annotation System (KOBAS) 2.0 (Xie et al., 2011) server was utilized to identify GO terms and KEGG pathways. The enrichment of each term was determined using the hypergeometric test and Benjamini–Hochberg FDR controlling procedure.

Other statistical analyses

Principal component analysis (PCA) analysis was performed by R package factoextra (https://cloud.r-project.org/package=factoextra) to show the clustering of samples with the first two components. The quantified experimental data of CCK-8 and EdU were presented as mean ± standard deviation (SD), and statistical analysis was performed using GraphPad Prism 9.0 software (GraphPad, La Jolla, CA, USA). Student’s t-test was used for comparison between two groups, and one-way ANOVA was used for comparison among multiple groups. A p-value <0.05 was considered statistically significant.

Results

Nintedanib inhibits the proliferation of gastric cancer cells

To determine whether nintedanib has an inhibitory effect on the growth of gastric cancer cells, AGS and MKN-28 gastric cancer cells were treated with different concentrations of nintedanib for 48 h, and cell viability was assessed using the CCK-8 assay. The results demonstrated that nintedanib inhibited the viability of both gastric cancer cell lines in a dose-dependent manner (Figs. 1A–1B). The results revealed that after 48 h of treatment with nintedanib, the IC50 value of AGS cells was 6.51 ± 0.49 μM, and the IC50 value of MKN-28 cells was 8.07 ± 0.13 μM. To visualize the effects of nintedanib on gastric cancer cell proliferation, we examined the changes in EdU-positive cells after treatment with different concentrations of nintedanib (1, 5, 10 μM) for 24 h. The EdU labeling showed a dose-dependent decrease in the percentage of EdU-positive cells compared to the control group (Figs. 1C–1F). Based on the observed strong inhibitory effect of nintedanib on the proliferation of gastric cancer cells, we further investigated its impact on apoptosis in these cells. The results indicated that after 48 h of nintedanib treatment, gastric cancer cells exhibited concentration-dependent apoptosis (Figs. 1G–1H). Taken together, these data indicate that nintedanib significantly inhibits the proliferation of gastric cancer cells in vitro and induces apoptosis.

Figure 1 Nintedanib inhibits the viability of gastric cancer cells.

(A–B) The effect of different concentrations of nintedanib on the viability of MKN-28 and AGS cells was assessed using the CCK-8 assay, and the absorbance values of the cells were measured at 48 h. (C–F) Cell proliferation activity of MKN-28 and AGS cells was determined using the EdU assay. (G–H) Flow cytometry was used to assess the apoptosis of AGS cells induced by nintedanib.Three independent experiments were conducted for each group. Scale bar: 100 µm. *P < 0.05,** P < 0.01,*** P < 0.001,**** P < 0.0001.

Identification of highly conserved regulated AS events responded to nintedanib therapy in gastric cancer cells

Log-phase AGS gastric cancer cells were incubated with 5 μM nintedanib for 48 h, and an untreated control group was also set up. RNA was extracted and subjected to RNA-Seq analysis. SUVA (Cheng et al., 2021) software was used to analyze the RNA-Seq data from the nintedanib-treated and control groups. We identified a significant number of splicing events in both datasets and determined the differential events among them (Fig. S1A). SUVA has reclassified the types of alternative splicing events (Fig. 2A). The number of five types of AS events identified by SUVA in RNA-Seq data and the number of RAS events analyzed for differences were counted. The main types of RAS events that occurred were alt5p and alt3p events (Figs. 2B and S1B). After mapping the differential splicing events identified by SUVA to classic splicing events, we found that A5SS, cassette exon, ES, and A3SS were the most frequently occurring alternative splicing events (Figs. 2C and S1C). Furthermore, approximately 80% of the AS events among these RAS events were complex splicing events (Figs. 2D and S1D), indicating the complexity of splicing events during nintedanib action in gastric cancer cells.

Figure 2 Alternative splicing event analysis during the inhibition of gastric cancer cells by nintedanib.

(A) The five types of splicing events detectable by the SUVA software. (B) The number of RAS events of each type identified in the transcriptome data by SUVA. (C) Splice junction constituting RAS events detected by SUVA was annotated to classical AS event types. And the number of each classical AS event types were showed with barplot. (D) D.The RAS number of the complex and simple RAS events. (E) Number of selected dominant splicing events in the RAS events (pSAR ≥ 50%, a total of 915 RAS events). (F) Principal component analysis (PCA) based on the splicing ratio values of RAS events with pSAR ≥ 50%. (G) Heatmap showing RAS events (pSAR ≥ 50%). (H) H.The top 10 most enriched GO terms (biological process) were illustrated for RAS (pSAR ≥ 50%) genes in the Drug vs Normal group. The colour scale showing the row-scaled significance (-log10 corrected P value) of the terms events with pSAR ≥ 50% (Fig. 2G). GO and KEGG pathway analyses were performed on the genes containing differentially spliced events between the two groups, and the enriched entries were visualized (Figs. 2H, S1D). Finally, we found that the DEGs were mainly related to the biological pathway of cell apoptosis.

Although there were relatively many splicing events, not all of them can function at the transcriptional level. The splicing rate pSAR represents the proportion of ASEs in the transcriptome of a gene relative to the entire transcriptome. We selected dominant splicing events (pSAR >= 50%) from the RAS events (915 RAS events in total) for further analysis (Fig. 2E). Using their splicing ratio values, we performed principal component analysis (PCA), which showed that the nintedanib treatment and control groups were separated along the first principal component (Fig. 2F). Heatmap visualization was used to show the selected RAS events with pSAR >= 50% (Fig. 2G). GO and KEGG pathway analyses were performed on the genes containing differentially spliced events between the two groups, and the enriched entries were visualized (Figs. 2H, S1E). Finally, we found that the DEGs were mainly related to the biological pathway of cell apoptosis. This aligns with our findings regarding the apoptosis induced by nintedanib, as presented in Fig. 1.

Alternative splicing genes involved in the apoptosis pathway

In the previous section of our results, we focused on the differential ASEs that were closely related to cell apoptosis between the two groups. Further studies were conducted on the RASEs in the cell apoptosis pathway, and it was found that there were seven variable splicing genes shared by the experimental and control groups. Among them, MAP3K10, BBC3, BAG6, NR4A1, and IFI27 were significantly upregulated after nintedanib treatment. The heatmap showed the ratio values of the relevant RASGs in the pathway (Fig. 3A). The splicing event of the gastric cancer drug resistance-related gene IFI27 significantly increased the selection proportion of the 3′end exon in the nintedanib treatment group (Fig. 3B). The selection proportion of the 5′end exon of the cell apoptosis-related gene BBC3 was significantly higher in the nintedanib treatment group than in the control group (Fig. 3C). Figures S2A–S2B show the differential expression levels and read distribution of the genes MAP3K10 and BAG6 in the two groups.

Figure 3 Differential expression analysis of apoptosis-related genes between the nintedanib group and the control group.

(A) Heat map displaying the ratio values of RASG in the apoptotic pathway. (B–C) Reads distribution diagram showing clualt3p17776 IFI27 and clualt5p27534 BBC3. Boxplot showing splicing ratio of clualt3p17776 IFI27 and clualt5p27534 BBC3 on the right. *: P value ≤ 0.05,**: P value ≤ 0.01,***: P value ≤ 0.001.

Construction of co-expression network between RNA binding proteins and RAS of genes involved in apoptotic process pathway

RBPs are typically identified as upstream regulatory factors in AS events during tumor development (Hentze et al., 2018). Differences in RBPs between the nintedanib-treated group and the control group may have led to the differential AS events currently being presented. We used DESeq2 to analyze all differentially expressed genes (FDR ≤ 0.01, FC ≥ 1.5 or ≤ 1/1.5) between the nintedanib-treated group and the control group (Fig. 4A). When we overlapped all differentially expressed genes with the currently known 2141 RBPs, we found that 134 differentially expressed RBPs (DERBPs) were involved in the process of nintedanib drug treatment (Fig. 4B). The heatmap showed that most of the differentially expressed RBPs were downregulated in the nintedanib group (Fig. 4C). GO functional enrichment analysis of all upregulated DERBPs between the two groups showed that these RBPs were significantly enriched in biological pathways related to type I interferon signaling pathway, innate immune response, and immune system process (Fig. S3A). Downregulated DERBPs were significantly enriched in biological pathways related to rRNA processing, ribosome biogenesis, tRNA processing, and tRNA modification (Fig. S3B). KEGG functional enrichment analysis showed that upregulated DERBPs were mainly enriched in biological pathways such as cell adhesion molecules (CAMs) and Inflammatory mediator regulation of TRP channels (Fig. S3C). The significantly enriched items in the KEGG analysis of downregulated DERBPs were spliceosome, ribosome biogenesis in eukaryotes, and the hippo signaling pathway, among others (Fig. S3D).

Figure 4 Differentially expressed RBPs may affect the expression of apoptosis-related genes during the inhibition of gastric cancer cells by nintedanib.

(A) Volcano plot was generated using DESeq2 to analyze all differentially expressed genes (FDR ≤ 0.01, FC ≥ 1.5 or ≤ 1/1.5) between Drug and Normal samples. (B) Venn diagram showing the overlap between all differentially expressed genes and 2141 currently known RBPs, identifying 134 differentially expressed RBPs (DERBPs) involved in the process of nintedanib drug treatment. (C) Heat map displaying the expression profiles of DERBPs in Drug and Normal groups. (D) Top 10 biological pathways from GO functional enrichment analysis of the RBP-RAS regulatory network. (E) Co-expression analysis of differentially expressed RBP and RAS events related to cell proliferation and angiogenesis (|Pearson correlation| ≥ 0.99, p-value ≤ 0.01) was performed to construct the RBP-RAS regulatory network. The identified RAS events and upstream differentially expressed RBPs were highlighted. GO functional enrichment analysis of the 134 differentially expressed DERBPs revealed significant enrichment in biological pathways related to apoptosis. (F) Reads distribution diagram showing clualt5p12044_NR4A1. Boxplot showing expression quantity of the RASG co-expressed DERBP of DHX9 and splicing ratio of clualt5p12044 NR4A1 on the right. *: P value ≤ 0.05,**: P value ≤ 0.01,***: P value ≤ 0.001.

To further explore the interaction relationship between DERBPs and differential AS events, we conducted co-expression analysis using the pSAR values of deRBP genes and relevant differential AS events. We identified typical DERBP genes that were co-expressed with differential AS events, extracted the genes where the co-expressed differential splicing events occurred, and conducted GO functional analysis. It was found that co-expressed genes were mainly enriched in pathways related to positive regulation of transcription by RNA (Fig. 4D). Co-expression analysis was also performed on differential RBPs and differential events of cell proliferation and angiogenesis-related genes that we focus on, and a regulatory network of RBP-RAS was constructed (Fig. 4E). It was found that the relevant genes included NOC2L, TAGLN, DHX9, and other abnormally expressed DERBPs, which could regulate the activity of downstream cell proliferation and angiogenesis-related genes such as NR4A1, BBC3, and IFI27. Box plots showed the expression levels of these DERBPs (Fig. S3E). We focused on the DHX9-NR4A1 relationship and presented the distribution plot and ratio value of DHX9 expression and clualt5p12044:NR4A1 reads (Fig. 4F).

Discussion

As is well known, the mechanism of action of drugs on tumor cells is often complex and network-like. Although nintedanib has been shown to have inhibitory effects on gastric cancer in multiple studies, current research is mostly limited to the study of single pathway mechanisms (Won et al., 2019; Zhu et al., 2023), which cannot meet our current needs. Therefore, this study first conducted RNA-seq analysis on gastric cancer cell samples treated with nintedanib and normal control group samples, in order to systematically explore the molecular processes and mechanisms by which nintedanib inhibits gastric cancer cells. Based on the RNA-seq data, we systematically studied the important RASEs and RBPs of gastric cancer cells treated with nintedanib at a whole-genome level.

Alternative splicing is closely related to gastric cancer, and previous studies have shown that selective splicing may involve multiple biological processes of gastric cancer. The selective splicing of CAV1 mediates the PTBP3-promoted metastasis of gastric cancer (Liang et al., 2018). Alternative splicing of PICALM regulated by long non-coding RNA can weaken the chemoresistance of gastric cancer (Zhang et al., 2021). Overexpression of lncRNA CCAT2 regulates the alternative splicing of CD44 variants, which can increase the transition from the standard form to the variable CD44v6 subtype, thereby promoting the proliferation, migration, and invasion of GC cells (Deng et al., 2023). An analysis of RNA sequence data from 348 GC samples showed that AS events (Exon Skip) in CLSTN1 and SEC16A may accelerate the biological progression of GC through the PI3K/AKT classic pathway (Feng et al., 2020). We conducted a GO enrichment analysis on the genes where AS events occurred in the transcriptome data of the nintedanib treatment group and the control group, and found that functional pathways are mainly enriched in transcriptional regulation, chromatin recombination, apoptosis, endocytosis, and decomposition metabolic pathways and other related biological pathways. Here we pay special attention to genes involved in the apoptosis pathway and related genes. Awasthi et al. (2021) have confirmed in preclinical models of gastric cancer that nintedanib can inhibit tumor cell proliferation, reduce tumor vascular system growth, and increase tumor cell death. This is consistent with our cell experimental results, so we believe that nintedanib may regulate the occurrence level of variable splicing genes related to cell apoptosis during the treatment of gastric cancer. We further analyzed the RASG involved in the process of nintedanib action on gastric cancer cells related to the apoptosis pathway, and found seven variable splicing genes, some of which are closely related to apoptosis and gastric cancer. BBC3 is a BH3 protein with strong pro-apoptotic ability, and the regulation of only bbc3′s transcription level can affect tumor cell survival (Han et al., 2001). Interferon α-inducible protein 27 (IFI27) overexpression can partially restore GC cell invasion, migration, and proliferation capacity (Li et al., 2022),while IFI-27 is overexpressed in the cisplatin-resistant cell line YCC-3/R and is a key gene involved in the process of tumor cell resistance (Lee et al., 2013). In summary, our study found that nintedanib can regulate cell apoptosis through post-transcriptional selective splicing pathway.

The post-transcriptional regulatory ability of RBPs enables them to influence RNA fate and is highly correlated with tumor occurrence and development (Hentze et al., 2018). Abnormal expression of RBPs has been shown to play an important role in various biological behaviors of gastric cancer and can be used to develop prediction models for gastric cancer patient prognosis (Abbas Raza et al., 2023; Ning et al., 2023). However, research on the variable splicing regulation of RBPs in gastric cancer is still in its early stages. A study on the RBP DEK in gastric cancer cells has shown that DEK affects the selective splicing of a large number of cancer-related genes in gastric cancer cells, which are significantly enriched in pathways including cell apoptosis and cell cycle processes (Liu et al., 2022). Polypyrimidine tract-binding protein 3 (PTBP3) increases the migration and invasion ability of gastric cancer cells through mediating CAV1 selective splicing (Liang et al., 2018). In this study, we found significant expression of 134 RBPs between two groups of samples, and most of them were downregulated after treatment with nintedanib, suggesting that nintedanib inhibits the functional expression of most RBPs. The co-expression network of RBPs and RASEs revealed that the network relationship formed by RBPs TAGLN2, TAGLN, SRSF6, PKM, SRSF2, NOC2L, IPO4, C1QBP, DHX9 is ultimately related to cell apoptosis. Tagln1 is usually upregulated in gastric cancer cells and activates the NRP2/VEGFR2 downstream MAPK signaling pathway, ultimately promoting angiogenesis (Jin et al., 2021). SRSF6 is an oncogene that may be overexpressed in most cancers. SRSF6 can regulate the alternative splicing of multiple genes, involving cell proliferation, migration, immune suppression, and drug resistance (She, Shao & Jia, 2021). NOC2L is considered an inhibitor of the histone acetyltransferase (INHAT) and a proto-oncogene, with increased expression in most cancer tissues including gastric cancer (Lu et al., 2023).

IPO4 is a histone transport protein that is usually highly expressed in gastric cancer tissues, and inhibiting IPO4 can weaken the proliferation and migration ability of gastric cancer cells (Xu et al., 2019). C1QBP is widely distributed in organelles and participates in various cellular processes including mRNA splicing. C1QBP plays a carcinogenic role in various tumors such as colon cancer, breast cancer, and lung cancer (Saha et al., 2019). DEAH box helicase 9 (DHX9), also known as nuclear DNA helicase II, has been shown to constitute a bidirectional regulatory mode in adenosine-to-inosine (A-to-I) editing, which is partially responsible for the dysregulation of editing spectrum in cancer. DHX9 may play an important role in inducing apoptosis in nintedanib-treated cells by regulating NR4A1 alternative splicing (Hong et al., 2018). NR4A1 has been confirmed to have anti-tumor effects in various tumors (Beard, Tenga & Chen, 2015; Wenzl et al., 2015; Wu et al., 2017), and research has shown that NR4A1 significantly enhances TNF α-induced GC cell apoptosis by inhibiting mitochondrial autophagy (Yan et al., 2018).

However, our study still has some limitations. We only selected a single gastric cancer cell line for analysis and lacked in vivo validation. We hope to validate the alternative splicing factors and differential RBPs in multiple gastric cancer cell lines and in vivo in subsequent experiments.

Conclusion

In this study, we used the SUVA software system to analyze RNA-seq data of nintedanib-treated gastric cancer cells. We identified 915 dominant transcriptional RAS events and 178 RBPs. Further functional enrichment analysis of RASEs revealed that these RASEs may be highly correlated with apoptosis in gastric cancer cells. The RBP-RASE regulatory network showed co-variation between RBPs and RASEs, suggesting a potential regulatory effect of RBPs on RASEs. This study revealed changes in the transcriptome of gastric cancer cells after nintedanib treatment, indicating that alternative splicing may affect the pro-apoptotic effects of nintedanib on gastric cancer cells under the regulation of RBPs.

Supplemental Information

Supplemental Information 1 Alternative splicing event analysis during the inhibition of gastric cancer cells by nintedanib

(A) A Venn diagram illustrating all splicing events and differential events detected in the normal control group and the drug treatment group. (B) Five types of AS events identified by SUVA include the predominant RAS event types, namely alt5p and alt3p events. (C) The bar chart displays the total number of splicing events identified by SUVA analysis and those obtained by conventional methods.(D) Mapping all splicing events identified by SUVA to classical splicing events, A5SS, cassette-Exon, ES, and A3SS are the most prevalent alternative splicing events.(E) KEGG enrichment analysis was conducted on genes associated with the splicing events depicted in fig1E.

Supplemental Information 2 Differential expression analysis of apoptosis-related genes between the nintedanib group and the control group

(A-B) The reads plot illustrates apoptosis-related genes: clualt5p26440:MAP3K10 and clualt5p43290:BAG6.

Supplemental Information 3 Differentially expressed RBPs may affect the expression of apoptosis-related genes during the inhibition of gastric cancer cells by nintedanib

(A) GO functional enrichment analysis was conducted on all upregulated DERBPs between Drug and Normal samples, which significantly enriched biological pathways such as type I interferon signaling pathway, innate immune response, and immune system process. (B) GO functional enrichment analysis was performed on all downregulated DERBPs between Drug and Normal samples, revealing significant enrichment in biological pathways including rRNA processing, ribosome biogenesis, tRNA processing, and tRNA modification. (C) KEGG functional enrichment analysis was carried out on upregulated DERBPs, revealing enrichment in biological pathways such as Cell adhesion molecules (CAMs) and Inflammatory mediator regulation of TRP channels. (D) KEGG functional enrichment analysis was conducted on downregulated DERBPs, showing enrichment in biological pathways such as Spliceosome, Ribosome biogenesis in eukaryotes, and Hippo signaling pathway. (E) The expression levels of DERBPs in fig3 E are as follows: TAGLN2, TAGLN, SRSF6, PKM, SRSF2, NOC2L, IPO4, C1QBP.

Additional Information and Declarations

Competing Interests

Author Contributions

DNA Deposition

Data Availability

The authors declare there are no competing interests.

Xiaohua Dong conceived and designed the experiments, performed the experiments, analyzed the data, prepared figures and/or tables, and approved the final draft.

Zhilong Liu conceived and designed the experiments, performed the experiments, analyzed the data, prepared figures and/or tables, and approved the final draft.

Miao Yu performed the experiments, prepared figures and/or tables, and approved the final draft.

Xiaojun Yang performed the experiments, authored or reviewed drafts of the article, and approved the final draft.

Hui Cai analyzed the data, authored or reviewed drafts of the article, and approved the final draft.

The following information was supplied regarding the deposition of DNA sequences:

The raw sequence data is available in the Genome Sequence Archive in National Genomics Data Center, China National Center for Bioinformation/Beijing Institute of Genomics, Chinese Academy of Sciences: GSA-Human: HRA0063970.

The following information was supplied regarding data availability:

The raw images and cell line experimental raw data are available at figshare: Dong, Xiaohua (2024). Identification of the whole genome of alternative splicing and RNA-binding proteins involved in nintedanib-induced apoptosis in gastric cancer cells. figshare. Figure. https://doi.org/10.6084/m9.figshare.26096158.v1.

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
