# Peer review of "Identification of the whole genome of alternative splicing and RNA-binding proteins involved in nintedanib-induced apoptosis in gastric cancer cells"

_PeerJ, doi:10.7717/peerj.18697_

## Round 0.1 · original submission · Major Revisions

Dear authors,

We kindly request that you carefully review the comments provided by the reviewers. Their valuable suggestions offer insights to enhance your manuscript. Incorporate their suggestions and carefully address all comments in your manuscript; it will significantly strengthen its content. Thanks

Reviewer 1 ·

Basic reporting

The paper is well written. Overall, the work is interesting and well-structured. The research questions are well-defined, relevant & meaningful.

Experimental design

I found the experimental section of the manuscript quite weak. It heavily relies on bioinformatic analysis and lacks experimental validation.

The manuscript demonstrates the involvement of alternative splicing events and RNA binding proteins as one reason for cells' apoptosis after nintedanib treatment. However, it does not include any data to support this. The authors should have included experimental validation, such as an apoptotic assay by flow cytometry or western blot, to demonstrate nintedanib-induced apoptosis.

The Methods section must provide sufficient detail and information to allow replication. I suggest the authors include the parameters for defining and quantifying alternative splicing and regulated alternative splicing events. Additionally, it would be useful to know how the data from each replicate were combined and whether the authors considered splicing events for the analysis if it is detected in only one replicate.

Validity of the findings

While reading the manuscript, I was hoping to see how the authors delve into the mechanisms and involvement of the RNA-binding proteins in apoptosis in gastric cancer. However, the manuscript was rather descriptive to the very end, relying entirely on the bioinformatic analysis.

I would suggest authors should validate their findings through RTqPCR or ddPCR.

Additional comments

In the second result section, I recommend that the authors explain the splicing events observed in untreated and drug-treated conditions. A Venn diagram showing the overlap and differences would be helpful.

In line 187, it's recommended to use "upregulated" instead of "activated," as no data supports increased gene activity.

·

Basic reporting

The article is written clearly and concisely. The flow of the article aligns with the outline.
The references included are adequate.
Overall, the article is written systematically, figures and tables are appropriate.

Experimental design

The objectives of the study are clearly defined. Experimental design and execution is in good agreement with the goals. Appropriate experiments as well as bioinformatics are applied.

Validity of the findings

The observations and results seems promising in understanding of impact of Nintedanib on proliferation of gastric cancer cells.

Additional comments

1. In the experimental design for proliferation assay (CCK-8), 6000 cells per well were selected. Was the study included in with other cell counts? I am curious especially to see the impact of nintedanib at lower concentration such as 1 µM or less.
2. In Figure 1A, how were the error bars measured? The error bar seems very low at higher concentrations?

---

## Round 0.2 · accepted · Accept

Authors conducted excellent workflow and explained role of RNA binding protein's and alternative splicing in Gastric cancer. This work shows more impact in gastric cancer research.

Reviewer 1 ·

Basic reporting

The authors addressed my comments.

Experimental design

Well planned

Validity of the findings

Already, comments are included in the first round of review

·

Basic reporting

Data reporting is clear and all the comments are addressed appropriately. Sufficient references are provided.

Experimental design

Experimental design and execution is appropriate, comments are addressed.

Validity of the findings

The findings are significant. Comments provided are addressed.

Additional comments

All the comments are addressed. I have no further comments.